# MEF2C Directly Interacts with Pre-miRNAs and Distinct RNPs to Post-Transcriptionally Regulate miR-23a-miR-27a-miR-24-2 microRNA Cluster Member Expression

**DOI:** 10.3390/ncrna10030032

**Published:** 2024-05-17

**Authors:** Estefanía Lozano-Velasco, Carlos Garcia-Padilla, Miguel Carmona-Garcia, Alba Gonzalez-Diaz, Angela Arequipa-Rendon, Amelia E. Aranega, Diego Franco

**Affiliations:** 1Cardiovascular Development Group, Department of Experimental Biology, University of Jaen, 23071 Jaen, Spain; evelasco@ujaen.es (E.L.-V.); cgpadill@ujaen.es (C.G.-P.); mcg00070@red.ujaen.es (M.C.-G.); aegd0001@red.ujaen.es (A.G.-D.); amar0007@red.ujaen.es (A.A.-R.); aaranega@ujaen.es (A.E.A.); 2Fundación Medina, 18016 Granada, Spain; 3Department of Anatomy, Embryology and Zoology, School of Medicine, University of Extremadura, 06006 Badajoz, Spain

**Keywords:** Mef2c, microRNAs, RNPs

## Abstract

Transcriptional regulation constitutes a key step in gene expression regulation. Myocyte enhancer factor 2C (MEF2C) is a transcription factor of the MADS box family involved in the early development of several cell types, including muscle cells. Over the last decade, a novel layer of complexity modulating gene regulation has emerged as non-coding RNAs have been identified, impacting both transcriptional and post-transcriptional regulation. microRNAs represent the most studied and abundantly expressed subtype of small non-coding RNAs, and their functional roles have been widely documented. On the other hand, our knowledge of the transcriptional and post-transcriptional regulatory mechanisms that drive microRNA expression is still incipient. We recently demonstrated that MEF2C is able to transactivate the long, but not short, regulatory element upstream of the miR-23a-miR-27a-miR-24-2 transcriptional start site. However, MEF2C over-expression and silencing, respectively, displayed distinct effects on each of the miR-23a-miR-27a-miR-24-2 mature cluster members without affecting pri-miRNA expression levels, thus supporting additional MEF2C-driven regulatory mechanisms. Within this study, we demonstrated a complex post-transcriptional regulatory mechanism directed by MEF2C in the regulation of miR-23a-miR-27a-miR-24-2 cluster members, distinctly involving different domains of the MEF2C transcription factor and the physical interaction with pre-miRNAs and Ksrp, HnRNPa3 and Ddx17 transcripts.

## 1. Introduction

Transcriptional regulation constitutes a key step in gene expression regulation. Multiple types of transcription factors have been identified from flies to humans, regulating multiple developmental, homeostatic and pathological processes [1,2,3]. In this context, a core of transcription factors has been identified to play essential roles in myogenesis, such as SRF, NKX2.5, GATA4 and MEF2C [4]. Myocyte enhancer factor 2C (MEF2C) is a transcription factor of the MADS box family involved in the early development of several cell types, including neural, immune, cartilaginous and endothelial cells, yet the main role of MEF2C is exerted by regulating muscle development (i.e., skeletal, cardiac and smooth) [5,6,7,8,9,10,11]. MEF2C-deficient mice are embryonically lethal, displaying complex cardiovascular defects as the early heart tube does not undergo looping morphogenesis, resulting in the absence of the future right ventricle [12,13]. Notably, MEF2C also plays a pivotal role in cardiac pathological conditions such as cardiac hypertrophy [14], and it represents an essential cornerstone for cardiac reprogramming [15,16].

Multiple studies have reported the essential role of MEF2C in regulating gene expression in different biological contexts, including cardiac [17,18,19,20,21], skeletal [7,22,23] and smooth muscle [24] cells. The transcriptional activity of MEF2C relies on its carboxyl terminal, a process that is sumoylation-dependent [25], whereas the MADS and MEF2 domains facilitate DNA-binding, dimerization and co-factor interactions [26,27,28]. Adjacent to the MEF2 domain is the HJURP-C (Holliday junction recognition protein C-termial) domain, followed by two transcriptional activation domains (TAD1 and TAD2), which are responsible for transcriptional activation [28]. Although different MEF2C isoforms have been reported, their functional role remains rather elusive [29].

Over the last decade, a novel layer of complexity in gene regulation has emerged with the identification of non-coding RNAs, impacting both transcriptional and post-transcriptional processes. Non-coding RNAs are broadly classified according to their transcript length into small non-coding RNAs (<200 nt) and long non-coding RNAs (>200 nt) [30]. Among the small non-coding RNAs, microRNAs represent the most studied and abundantly expressed subtype. MicroRNAs are nuclearly encoded and transcribed into microRNA precursor molecules by RNA polymerase II. In certain genomic locations, microRNAs are clustered, resulting in a primary transcript containing multiple microRNA precursors, leading to a pri-mRNA precursor. Pri-miRNAs are then processed by nucleases such as Drosha and Dgcr8 to generate distinct pre-miRNA molecules that are subsequently exported by the exportin-5/Ran protein complex to the cytoplasm [31]. Within the cytoplasm, the pre-miRNAs are further processed into mature microRNA duplex by Dicer RNAse and loaded into the RISC complex. Within the RISC complex, one strand of the double-stranded microRNA molecule is degraded, leaving the mature microRNA to scan RNA molecules for the sequence homology of its seed sequence, leading to post-transcriptional RNA target cleavage, translation repression and/or RNA deadenylation. As a result, in most cases, the abundance of the miRNA/protein target decreases [32]. Importantly, emerging evidence suggests that certain microRNAs can also modulate transcriptional regulation by exerting their function within the nucleus, thus impacting alternative splicing and RNA and microRNA transcriptional regulation [33].

MicroRNAs are highly conserved throughout evolution, ranging from *C. elegans* to humans. MicroRNAs display temporal and spatial differential expression during embryonic and adulthood, thereby contributing to both embryonic development and tissue homeostasis [34]. Impaired expression and/or function of microRNAs have thus been recently reported to lead to pathological conditions [35,36,37,38,39]. In this context, several microRNAs have been reported to be regulated by MEF2C [40], while conversely, several microRNAs also modulate MEF2C expression in different biological contexts [41,42,43,44,45,46,47].

The functional role of microRNAs during cardiac development and diseases has been extensively documented [35,36,37,38,48]. Within this context, the genetic deletion of miR-1-2 [49,50] and miR-126 [51,52], respectively, has revealed the essential role of these miRNAs in cardiac and vascular embryonic development. Importantly, several clustered microRNAs, such as miR-19-72 [53,54,55], miR-106a-363 [56], miR-106b-25 [57,58] and miR-23/27/24 [59,60,61], have been reported to play modulatory roles in distinct biological contexts, including the cardiovascular system [58,59,62,63,64]. While our current understanding of the detailed functional relevance of each of these microRNAs is progressively emerging, our knowledge of the transcriptional and post-transcriptional regulatory mechanisms that drive microRNA expression is still incipient.

Lee et al. [65] demonstrated that the miR-23a-miR-27a-miR-24-2 cluster is transcribed as an RNA polymerase II-dependent primary transcript whose main transcriptional regulation is driven by a −600 bp upstream promoter. We subsequently reported the identification of upstream regulatory elements driving miR-23a-miR-27a-miR-24-2 transcriptional regulation in both cardiac and skeletal muscle cells [66]. Within this context, we demonstrated that MEF2C is able to transactivate the long (−1830 to +1 nt) regulatory element but not the short (−776 to +1 nt) element, in accordance with the identification of MEF2 regulatory binding sites distribution. However, MEF2C over-expression and silencing, respectively, displayed distinct effects on each of the miR-23a-miR-27a-miR-24-2 cluster members without affecting pri-miRNA expression levels in different cell types [66], thus supporting additional MEF2C-driven regulatory mechanisms. Within this study, we report complex transcriptional and post-transcriptional regulatory mechanisms directed by MEF2C in the regulation of the miR-23a-miR-27a-miR-24-2 cluster, distinctly involving different domains of the MEF2C transcription factor and the physical interaction with pre-miRNAs and Ksrp, HnRNPa3 and Ddx17 transcripts.

## 2. Results

We have previously characterized the transcriptional potential of the 1.8 Kb upstream sequences of the miR-23a-miR-27a-miR-24-2 cluster and reported that MEF2C is capable of transcriptionally activating these regulatory regions in HL1 atrial cardiomyocytes. Such transcriptional activation thus enhances the expression of the miR-23a-miR-27a-miR-24-2 pri-miRNA. However, we have also previously reported that MEF2C over-expression and inhibition, respectively, distinctly regulate the expression of each of the miR-23a-miR-27a-miR-24-2 mature cluster members independently of its transcriptional potential. Notably, the modulation of miR-23a-miR-27a-miR-24-2 cluster members by MEF2C is tissue-specific. Therefore, our previous data suggest that MEF2C modulation of miR-23a-miR-27a-miR-24-2 cluster members is exerted by either direct or indirect post-transcriptional mechanisms. Since microRNAs have been recently reported to exert both cytoplasmic and nuclear functions, we initially explored the subcellular location of the miR-23a-miR-27a-miR-24-2 mature cluster members. RT-qPCR of nuclear and cytoplasmic fractions revealed that all three members, i.e., miR-23a_3p, miR-27a_3p and miR-24_3p, are similarly localized in both subcellular compartments in HL1 cardiomyocytes (Figure 1A), in contrast to miR-130a, which is preferentially and significantly enhanced in the cytoplasm (Figure 1A), while Xist2 is enhanced in the nucleus (Figure 1A), serving as internal subcellular fractioning control.

Subsequently, we therefore tested whether MEF2C modulation of miR-23a-miR-27a-miR-24-2 cluster members is exerted by either direct or indirect post-transcriptional mechanisms. We first explored the plausible interaction between MEF2C and miR-23a-miR-27a-miR-24-2 cluster members by performing MEF2C protein pulldown assays (Appendix A). Our data demonstrated that MEF2C interacts with pre-miR-23a and pre-miR-27a but not with pre-miR-24-2 (Figure 1B). Importantly, MEF2C neither binds to the mature microRNAs, i.e., miR-23a_3p, miR-27a_3p and miR-24_3p (Figure 1C) nor the miR-23a-miR-27a-miR-24-2 pri-miRNA (Figure 1D), demonstrating a direct post-transcriptional role of MEF2C governing the expression of the precursor forms of miR-23a_3p and miR-27a_3p but not miR-24_3p (Figure 1E).

We subsequently tested which part of the MEF2C transcription factor exerts pre-miR-23a and pre-miR-27a modulation and whether it also affects the expression of the mature microRNA cluster members. For this purpose, we constructed two distinct MEF2C variants. The first one lacks the SRF-type DNA-binding and dimerization domain, the MADS_MEF2_like and the HJURP_C domain at the 5’ end (MEF2C 5′del) (Appendix A). The second variant lacks the 3’ end (MEF2C 3′del) while maintaining these domains (Appendix A). In addition, we also performed MEF2C over-expression and silencing studies, achieving successful modification of MEF2C expression levels. In addition, the over-expression of MEF2C full-length, MEF2C 5′del and MEF2C 3′del showed increased levels, both at the transcript and protein levels, as compared to non-transfected controls, respectively (Appendix A). Furthermore, transactivation assays of the L regulatory element of the miR-23a-miR-27a-miR-24-2 locus were successfully achieved with MEF2C full-length and MEF2C 3′del constructs but not with MEF2C 5′del, as expected, since the latter lacks DNA-binding and dimerization domains (Appendix A).

RT-qPCR analyses of precursor pre-miR-23a, pre-miR-27a and pre-miR-24-2, followed by Sanger sequencing (Appendix A), demonstrate that both MEF2C 5′del and MEF2C 3′del significantly increase the steady-state levels of pre-miR-23a_3p and pre-miR-27a_3p but not of pre-miR-24_3p (Figure 2A), while the over-expression of full-length MEF2C and silencing resulted in minimal but significant downregulation of pre-miR-23a and pre-miR-27a but not of pre-miR-24-2 (Figure 2A). Thus, these data reinforced the previous observations by MEF2C pulldown experiments, demonstrating a regulatory role on pre-miR-23a and pre-miR-27a but not on pre-miR-24-2. RT-qPCR analyses of mature miR-23a-miR-27a-miR-24-2 cluster members demonstrate that MEF2C full-length over-expression significantly downregulates miR-23a_3p and miR-27a_3p but not miR-24_3p expression (Figure 2B), while MEF2C silencing selectively downregulates only miR-27a_3p. Importantly, MEF2C 3′del and MEF2C 5′del significantly upregulate all mature microRNA cluster members, i.e., miR-23a_3p, miR-27a_3p and miR-24_3p (Figure 2B).

Therefore, our data indicate that both the 5’ end and 3’ end of MEF2C regions play inhibitory roles in modulating mature miR-23a-miR-27a-miR-24-2 cluster members. Moreover, mature miR-24_3p expression is modulated by both MEF2C 5′del and MEF2C 3′del but not its precursor form (pre-miR-24-2) (Figure 2C), suggesting a dual role for MEF2C in regulating such post-transcriptional events (Figure 2A,B). However, it should be taken into account that miR-24-1 precursor, located in a distinct chromosomal locus, might also contribute to miR-24_3p expression levels.

Modulation of miR-23a-miR-27a-miR-24-2 cluster members by MEF2C can also be indirectly exerted through association with ribonucleic proteins (RNPs). We therefore tested the tissue-specific expression of distinct RNPs previously reported to interact with distinct microRNAs in different cell types [67,68,69,70]. We evaluated the expression of seven distinct RNPs (Adar1, Ddx5, Ddx17, HnRNPa1, HnRNPa3, HnRNPa2b1 and Ksrp) in 3T3 fibroblasts, HL1 atrial cardiomyocytes and Sol8 skeletal myoblasts.

Adar1, Ddx5 and HnRNPa1 displayed increased expression in 3T3 fibroblasts as compared to HL1 atrial cardiomyocytes and Sol8 skeletal myoblasts (Figure 3A). Ksrp displayed similarly enhanced expression in 3T3 fibroblasts and HL1 atrial cardiomyocytes as compared to Sol8 skeletal myoblasts (Figure 3A). HnRNPa2b1 and Ddx17 display a similar expression pattern with enhanced expression in HL1 cardiomyocytes, while HhRNPa3 displayed the opposite pattern, i.e., decreased expression in HL1 atrial cardiomyocytes as compared to 3T3 fibroblasts and Sol8 skeletal myoblasts (Figure 3A). Overall, these data showed that the RNA constituents of all mentioned RNPs are expressed in these three distinct cell lines tested. However, the distinct RNPs showed a differential expression in these cell lines, thus supporting the plausible contribution of these RNPs in regulating the distinct miR-23a-miR-27a-miR-24-2 cluster members by MEF2C in different cell types, as previously demonstrated [64].

We also tested whether these RNP transcripts are distinctly distributed within the subcellular compartments in HL1 cardiomyocytes. Our data revealed that Adar1 is highly enriched in the nuclear compartment, whereas Ddx5, Ddx17 and Ksrp are prominently localized in the cytoplasm. On the other hand, HnRNPa1, HnRNPa3 and HnRNPa2b1 are similarly distributed within both nuclear and cytoplasmic compartments, in line with MEF2C mRNA distribution (Figure 3B).

We additionally tested whether these RNP transcripts are regulated by MEF2C. Over-expression of MEF2C full-length resulted in the upregulation of Adar1 and downregulation of HnRNPa3 and Ksrp, while Ddx5, Ddx17, HnRNPa1 and HnRNPa2b1 were not altered (Figure 3C). MEF2C silencing decreased Ddx17, HnRNPa3, HnRNPa2b1 and Ksrp while increasing HnRNPa1. Adar1 and Ddx5 did not display significant differences (Figure 3C). MEF2C 5′del significantly increased Adar1 and significantly downregulated Ddx5, Ddx17, HnRNPa1, HnRNPa3, HnRNPa2b1 and Ksrp (Figure 3C), while MEF2C 3′del significantly increased Adar1, HnRNPa1 and HnRNPa3, while Ddx5, Ddx17 and Ksrp displayed significant downregulation (Figure 3C). HnRNPa2b1 displayed no significant differences after MEF2C 3′del over-expression (Figure 3C). In sum, our data demonstrate that these RNP transcripts are distinctly modulated by MEF2C. Particularly, it is important to highlight that Ksrp is similarly downregulated in all experimental conditions, Ddx17 is downregulated in MEF2C 3′del, MEF2C 5′del and MEF2C siRNAs conditions, while HnRNPa3 is downregulated by MEF2C full-length, MEF2C siRNA and MEF2C 3′del. Furthermore, Ddx5, Ddx17 and Ksrp are downregulated by MEF2C 3′del and MEF2C 5′del over-expression (Figure 3D), supporting the plausible role of these RNPs in the distinct regulation of miR-23a-miR-27a-miR-24-2 cluster members and/or its precursors.

We subsequently tested whether MEF2C can interact with these RNP transcripts. MEF2C protein pulldown demonstrated that Ddx17, Ksrp and HnRNPa3 transcripts interacted with MEF2C (Figure 4A), while no significant interaction was observed for HnRNPa1, HnRNPa2b1, Ddx5 and Adar1, respectively (Figure 4A). Therefore, these data demonstrate that MEF2C can directly interact with distinct RNP transcripts and thus further support the notion that MEF2C can post-transcriptionally modulate additional RNA transcripts involved in miR-23a-miR-27a-miR-24-2 cluster member expression.

Finally, we performed RNPs silencing assays in order to determine whether RNP inhibition can modulate miR-23a-miR-27a-miR-24-2 cluster members. Ddx17, HnRNPa1 and HnRNPa2b1 siRNA administration did not significantly modulate miR-23a_3p, miR-27a_3p and miR-24_3p pre-miRNAs, respectively, except for HnRNPa1, which significantly upregulated pre-miR-27a and pre-miR-24-2 (Figure 4B,D). On the other hand, Ddx5, HnRNPa3 and Ksrp inhibition enhanced miR-23a_3p, miR-27a_3p and miR-24_3p pre-miRNAs, except for pre-miR-24-2 after Ksrp inhibition (Figure 4B,D). For the mature miR-23a-miR-27a-miR-24-2 cluster members, Ddx17 and HnRNPa1 silencing significantly upregulated, while Ddx5 significantly downregulated all miR-23a-miR-27a-miR-24-2 cluster members, i.e., miR-23a_3p, miR-27a_3p and miR-24_3p (Figure 4C,D). On the other hand, HnRNPa2b1 inhibition exclusively upregulated miR-23a_3p, but not miR-27a_3p and miR-24_3p, while HnRNPa3 silencing led to downregulation of miR-23a_3p, upregulation of miR-27a_3p and no significant modulation of miR-24_3p (Figure 4C,D). Finally, Ksrp silencing led to downregulation of all miR-23a-miR-27a-miR-24-2 cluster members, i.e., miR-23a_3p, miR-27a_3p and miR-24_3p (Figure 4C,D). Thus, these observations revealed that Ddx5, HnRNPa3 and Ksrp are essential primarily for pre-miR-23a and pre-miR-27a and, to a lesser extent, for pre-miR-24-2 (only Ddx5 and HnRNPa3), supporting a key role modulating differential expression of the miR-23a-miR-27a-miR-24-2 cluster members. Similarly, HnRNPa2b1, HnRNPa3 and Ksrp silencing also distinctly modulate mature miR-23a-miR-27a-miR-24-2 cluster members. In sum, these data illustrate that distinct RNPs can impact differential pre-miRNA and mature miR-23a-miR-27a-miR-24-2 cluster member expression.

## 3. Discussion

Within the last decade, our understanding of the functional role of distinct microRNAs has greatly emerged; however, our knowledge of the transcriptional and post-transcriptional regulatory mechanisms driving microRNA expression is still incipient. We previously demonstrated that MEF2C over-expression and silencing, respectively, displayed distinct effects on each of the mature miR-23a-miR-27a-miR-24-2 cluster members [66], thus supporting additional MEF2C-driven regulatory mechanisms. We provide herein evidence that MEF2C can directly bind to pre-miR23a and pre-miR-27a but not to pre-miR-24-2. Importantly, MEF2C does not directly bind to either the pri-miRNA miR-23a-miR-27a-24-2 precursor or to the mature miR-23a_3p, miR-27a_3p and miR-24_3p molecules. Furthermore, we also demonstrated that distinct MEF2C domains can differentially modulate both pre-miRNA and microRNA expression. While there is emerging evidence that distinct proteins can influence MEF2C expression levels, leading to sumoylation and caspase cleavage [26,71], this is, to the best of our knowledge, the first proof that a transcription factor can influence microRNA biogenesis by directly interacting with pre-miRNA molecules.

On the other hand, ample evidence is reported on the key role of distinct ribonucleoproteins (RNPs) in modulating microRNA expression [72,73,74,75,76,77]. Thus, to further support the plausible role of several of these RNPs in MEF2C-driven miR-23a-miR-27a-miR-24-2 expression, we analyzed the expression of seven distinct RNP transcripts in three distinct cell types (fibroblasts, cardiomyocytes and skeletal muscle myoblasts), demonstrating that all of them are indeed expressed while displaying cell type enrichment, i.e., Ddx17 and HnRNPa2b1 are more abundantly expressed in cardiomyocytes, while Adar1, Ddx5 and HnRNPa3 are widely expressed in fibroblasts. Furthermore, we demonstrated that these RNP transcripts displayed distinct subcellular distribution patterns, i.e., Adar1 is primarily located in the nucleus, Ddx5, Ddx17 and Ksrp are primarily in the cytoplasm, while HnRNPa1, HnRNPa2b1 and HnRNPa3 are both nuclear and cytoplasmic, in line with previous reports [78,79,80,81,82,83]. Importantly, we firstly demonstrated that mature miR-23a-miR-27a-miR-24-2 cluster microRNA members are equally distributed in both nuclear and cytoplasmic subcellular compartments, supporting the notion that they might exert distinct functional roles, as recently reported [84,85,86,87,88], and thus can be distinctly regulated in the cytoplasm vs. the nucleus. Furthermore, these data also support that distinct RNPs might be involved in the differential and subcellular compartment-specific expression of miR-23a-miR-27a-miR-24-2 cluster members.

Scarce evidence has been reported for transcription factors directly binding to RNPs [88], supporting their plausible role in post-transcriptional regulation. For MEF2C, only AUF1 binding has been reported, promoting skeletal muscle myogenesis [83]. Within this study, we report for the first time that MEF2C can directly bind to Ddx17, HnRNPa3 and Ksrp mRNAs, respectively. Additionally, MEF2C indirectly regulates Adar1 and HnRNPa2b1 expression. Furthermore, we also demonstrate that distinct MEF2C domains differently contribute to RNP transcript expression. In this context, both 5′ and 3’ MEF2C ends can selectively inhibit Ddx5, Ddx17 and Ksrp expression while enhancing Adar1 expression. On the other hand, HnRNPa1, HnRNPa3 and HnRNPa2b1 are distinctly regulated by MEF2C 3′ and 5’ ends, respectively. While additional studies are required to fully understand the molecular mechanisms directing MEF2C 3′ and 5’ ends modulation of these RNPs, our data support the notion that they might be transcriptionally regulated since the MEF2C 5′del construct lacks transcriptional potential (Appendix A) and primarily downregulates their expression, while the MEF2C 3′del construct displays the opposite pattern. In sum, our data demonstrate that MEF2C can directly and indirectly regulate distinct RNPs in cardiomyocytes, with a potential impact on miR-23a-miR-27a-miR-24-2 cluster member expression.

As previously stated, a large body of evidence has been reported on RNPs modulating microRNA expression [75,76,77,78,79,80,89,90,91,92], yet their role in differential microRNA cluster expression has only been reported for Adar1 [84,85,86,87] and Ksrp [68,93,94]. Several studies reported miR-27b regulating Ksrp expression in distinct biological settings [95,96,97], but no proof has been reported for miR-27b being regulated by Ksrp. miR-27b regulation by RNPs has only been reported for HnRNPa1 in colorectal cancer [98] and HnRNPa2b1 in preeclampsia [99]. Importantly, no data have been reported for the involvement of these RNPs in the regulation of the miR-23a_3p or miR-24, as well as for any of the miR-23a-miR-27a-miR-24-2 cluster members, except for Ksrp regulating miR-23a_3p [100] and Adar1 regulating pre-miR-27a_3p to mature miR-27a_3p processing in cancer [101].

We provide herein evidence that silencing Krsp selectively upregulates pre-miR-23a and pre-miR-27a but not pre-miR-24-2 expression. Similarly, silencing HnRNPa1 leads to pre-miR-27a and pre-miR-24-2 downregulation without affecting pre-miR-23a expression. For the mature microRNAs, Ksrp and Ddx5 inhibition diminished all mature miR-23a-miR-27a-miR-24-2 cluster members, while selective inhibition of Ddx17 and HhRNPa1 enhanced all mature miR-23a-miR-27a-miR-24-2 cluster members. Curiously, HnRNPab1 silencing selectively upregulates miR-23a_3p but not miR-27a_3p and miR-24_3p, while HnRNPa3 silencing upregulates miR-23a_3p and downregulates miR-27a_3p and miR-24_3p. It is important, nonetheless, in this context that miR-24 levels might result from the amplification of the mature miR-24_3p from both pre-miR-24-1 and pre-miR-24-2 precursors, as previously mentioned. Importantly, MEF2C directly interacts with Ddx17, HnRNPa3 and Ksrp, and MEF2C silencing is essential for proper Ddx17, HnRNPa1, HnRNPa2b1, HnRNPa3 and Ksrp, also proving proof of the differential role of the MEF2C C-terminal and N-terminal in this regulation.

Overall, these data demonstrate the complex and pivotal role of distinct RNPs in regulating miR-23a-miR-27a-miR-24-2 cluster members and support the notion that distinct RNPs, particularly HnRNPa1 and Ksrp, play a pivotal role in regulating the differential expression of miR-23a-miR-27a-miR-24-2 cluster members by selectively acting on distinct pre-miRNAs. Surprisingly, the selective inhibition of mature miR-23a-miR-27a-miR-24-2 cluster members by RNP silencing is observed only for HnRNPa2b1 and HnRNPa3, but they do not recapitulate the effects provided by MEF2C silencing, supporting the notion that combinatorial rather than single MEF2C-driven RNP modulation is occurring. Furthermore, it is important to highlight in this context that all mature miR-23a-miR-27a-miR-24-2 cluster members are similarly expressed in both subcellular nuclear and cytoplasmic compartments as well as several RNPs. Notably, siRNA silencing would only be affecting those events occurring in the cytoplasm, and therefore, inhibition might only be partial. The causal relationship between such distinct subcellular compartment localization deserves further analysis and might provide novel insights into the precise molecular mechanisms controlling the differential expression of the mature microRNAs of genomic clustered microRNAs.

In summary, we provide herein evidence of the complex post-transcriptional regulatory mechanism exerted by MEF2C in the regulation of miR-23a-miR-27a-miR-24-2 cluster members (Figure 5). MEF2C can directly and selectively bind to pre-miR-23a_3p and pre-miR-27a_3p but not to pre-miR-24-2. Additionally, MEF2C can directly bind to distinct RNP transcripts, such as Ddx7, HhRNPa3 and Ksrp, while indirectly regulating the expression of other RNPs, such as Adar1 and HnRNPa2b1. Importantly, such regulation is distinctly exerted by the MEF2C amino- and carboxy-terminals. Silencing of MEF2C-binding RNP Ksrp selectively regulates pre-miR-23a and pre-miR-27a expression but not pre-miR-24-2, supporting the notion of a direct implication of this pathway on the differential expression of miR-23a-miR-27a-miR-24-2 cluster members, yet a combinatorial action of distinct RNPs seems to be required to fully achieve the final miR-23a-miR-27a-miR-24-2 cluster member expression of the mature microRNAs.

## 4. Materials and Methods

### 4.1. MEF2C Pulldown Assays

For the immunoprecipitation of endogenous MEF2C, protein A-Sepharose beads (Abcam, Cambridge, UK) were coated with 15 μg of antibody that recognized MEF2C (#9792-Cell Signalling) or control IgG (Abcam, Cambridge, UK) overnight at 4 °C with rotation. The next day, HL1 cells were lysed with PEB buffer (100 mM KCl, 5 mM MgCl_2,_ 10 mM Hepes, pH 7.0, 0.5% Nonidet P-40, 1 mM DTT, 100 units/mL RiboLOCK and Complete Protease Inhibitor Cocktail) for 10 min on ice and centrifuged at 10,000× *g* for 30 min at 4 °C. The supernatants were incubated with previously mentioned protein A-Sepharose-coated beads with 15 μg of antibody that recognized MEF2C or control IgG for 2 h at 4 °C with rotation, respectively. The corresponding beads were washed with NT2 buffer (50 mM Tris–HCl [Ph 7.5], 150 mM NaCl, 1 mM MgCl2, 0.05% NP-40) two times after spinning down at 5000 g for 2 min at 4 °C. Protein complexes were incubated with 20 units of DNase I (15 min at 37 °C). In this step, an aliquot from each reaction was isolated for Western blot validation. Subsequently, they were further incubated with 0.1% SDS/0.5 mg/mL Proteinase K (30 min at 55 °C) with mixing to remove DNA and proteins, respectively, and centrifuged at 5000× *g* for 5 min to collect the supernatant. The RNA isolated from the IP materials (acid phenol-chloroform) was further assessed by RT-qRT-PCR analysis.

### 4.2. Nuclear/Cytoplasmic Distribution

Cytoplasmic and nuclear RNA fractions from HL1 cardiomyocytes were isolated with a Cytoplasmic & Nuclear RNA Purification Kit (Norgen, Belmont, CA, USA) following the manufacturer’s instructions. After RNA isolation, RT-qPCR analysis for nuclear enriched *Xist2* mRNA marker and cytoplasmic *Gapdh* mRNA marker were performed to validate enrichment on each subcellular fraction. RT-qPCR analysis of distinct microRNAs, *Xist2* and RNPs was subsequently performed, as detailed in the next sections.

### 4.3. Generation of MEF2C 3′ Deletion and 5′ Deletion Constructs

The pcDNA MEF2C plasmids were used to generate two distinct constructs, with 3′ and 5′ deletions, respectively [102]. MEF2C 3′ deletion (MEF2C 3′del) was constructed by deletion of the 3′ fragment ranging from nucleotide 1112 of the MEF2C full-length (PstI restriction site) to the 3’ end of the mouse MEF2C transcript (NM_001170537.2) (Appendix A). Thus, this construct deleted the last 288 amino acids of the MEF2C protein (NP_001164008), thus maintaining the SRF-type DNA-binding and dimerization domain (1–59 aa), the MADS_MEF2_like domain (2–78 aa) and the HJURP_C domain (110–156 aa).

MEF2C 5′ deletion (MEF2C 5′del) was constructed by deletion of the 5′ fragment spanning from nucleotide 1 until nucleotide 1522 of the MEF2C full-length (ScaI restriction site), i.e., mouse MEF2C transcript (NM_001170537.2) (Appendix A). Thus, this construct deleted the first 314 amino acids of the MEF2C protein (NP_001164008), thus deleting the SRF-type DNA-binding and dimerization domain (1–59 aa), the MADS_MEF2_like domain (2–78 aa) and the HJURP_C domain (110–156 aa).

### 4.4. Plasmid Transfections

HL1 cardiomyocytes (6 × 10^4^ cells per well) were transfected with a plasmid containing MEF2C open reading frame (ORF) full-length (wt), MEF2C 5′ deletion (MEF2C 5′del), MEF2C 3′ deletion (MEF2C 3′del) at plasmid concentration 400 ng per well using lipofectamine 2000 (Invitrogen, Waltham, MA, USA) according to the manufacturer’s guidelines and incubated at 37 °C for 24 h, as previously described [100,101,102].

### 4.5. siRNA Transfections

HL1 cardiomyocytes (6 × 10^5^ cells per well) were transfected with siRNA-MEF2C, siRNA-Adar1, siRNA-HnRNPa3, siRNA-Ksrp, siRNA-Ddx5, siRNA-Ddx17, siRNA-HnRNPa1 and siRNA-HnRNPa2b1 (Sigma, Aldrich, Munich, Germany), respectively, at a siRNA concentration of 40 nM per well using lipofectamine 2000 (Invitrogen, Waltham, MA, USA) according to the manufacturer’s guidelines and incubated at 37 °C for 48 h, as previously described [98,99,100]. siRNA sequences are provided in Appendix A.

### 4.6. RNA Isolation and Retrotranscription

Total RNA was isolated using the ReliaPrep RNA Cell Miniprep System (Promega, Madison, WI, USA), and DNase was treated using RNase-Free DNase according to the manufacturer’s guidelines for 15 min at room temperature. In all cases, at least three distinct pooled samples were used to perform the corresponding RT-qPCR experiments.

### 4.7. RT-qPCR Analyses (mRNA)

First-strand cDNA was synthesized by using 100 ng of total RNA and a reverse transcription Maxima First Strand cDNA Synthesis Kit for RT-qPCR (Thermo Scientific, Waltham, MA, USA) according to the manufacturer’s guidelines. Negative controls to assess genomic contamination were performed for each sample, without reverse transcriptase, which resulted in all cases in no detectable amplification product. Real-time PCR experiments were performed with 2 μL of diluted cDNA, GoTaq qPCR Master Mix (Promega) and corresponding primer sets. Two internal controls, mouse *Gusb* and *Gapdh* mRNAs, were used in parallel for each run and represented as previously described [103,104,105]. Amplification conditions were as follows: denaturalization step of 95 °C for 10 min, followed by 40 cycles of 95 °C for 30 s, 60 °C for 30 s and 72 °C for 30 s, with a final elongation step of 72 °C for 10 min. All primers were designed to span exon–exon boundaries using the online Primer3 software Primer3input (http://bioinfo.ut.ee/primer3-0.4.0/, accessed on 12 January 2022). Primer sequences are provided in Appendix A. Amplification bands of pri-miRNA and pre-miRNAs are illustrated in Appendix A, demonstrating a single transcript for pri-miRNA miR-23a-miR-27a_miR-24-2 and specific amplifications for each pre-miRNA, i.e., pre-miRNA-23a, pre-miR-27a and pre-miR-24-2, respectively. No amplifications were observed in PCR control reactions containing only water as a template. Each PCR reaction was performed at least three times to obtain representative averages. The Livak method was used to analyze the relative quantification RT-qPCR data [106] and normalized in all cases, taking 100% as the wild-type (control) value, using *Gapdh* and *Gusb* as internal control for mRNA expression analyses, as previously described [103,104,105].

### 4.8. qRT-PCR Analyses (microRNA)

For microRNA expression analyses, 20 ng of total RNA was used for retrotranscription with a Universal cDNA Synthesis Kit II (Exiqon, Venlo, The Netherlands), and the resulting cDNA was diluted 1/80, following the manufacturer’s guidelines. Real-time PCR experiments were performed with 1 μL of cDNA, GoTaq qPCR Master Mix (Promega) and corresponding primer sets, as described in Appendix A. All RT-qPCRs were performed using a CFX384TM thermocycler (Bio-Rad, Hercules, CA, USA) following the manufacturer’s recommendations. The relative level of expression of each gene was calculated as described by Livak and Schmittgen [106] using *5S* as an internal control for microRNA expression analyses. Each PCR reaction was performed at least three times to obtain representative averages.

### 4.9. Western Blot

Western blot was performed using 30 μg of total protein. The primary antibodies Mef2c (sc-13268; Santa Cruz Biotechnology, Dallas, TX, USA) and Tubulin (sc-8035; Santa Cruz Biotechnology, Dallas, TX, USA) were used at a concentration of 1:100 and 1:5000, respectively, and incubated overnight at 4 °C and the secondary antibody-HRP conjugate (#170-6516, Biorad, Hercules, CA, USA) at 1/5000 for 2 h at room temperature. Blocking was carried out with 5% milk and washed with PBST according to the antibody manufacturer’s recommendations.

### 4.10. Luciferase Assay

Promotor distal sequence (L) was amplified from mouse genomic DNA with specific primers bearing HindIII/BamHI restriction sites and cloned into a pGLuc-Basic vector (New England Biolabs, Ipswich, MA, USA). 3T3 fibroblasts (ATCC, Manassas, VA, USA) were co-transfected with 100 ng of the L-pGluc vector, 300 ng of pcLux vector control for internal normalization and 400 ng from Mef2c FL, Mef2c 3′ or Mef2c 5′, respectively. Luciferase activity was measured 24 h after transfection by using the Pierce Gaussia Luciferase Flash Assay Kit (Thermo Fisher Scientific, Rockford, IL, USA) and normalized to pcLux vector control by using the Pierce Cypridina Luciferase Flash Assay Kit (Thermo Fisher Scientific, Rockford, IL, USA). In all assays, transfections were carried out in triplicate.

### 4.11. Statistical Analyses

For statistical analyses of datasets, unpaired Student’s *t*-tests were used. Significance levels or *p* values are stated in each corresponding figure legend. *p* < 0.05 was considered statistically significant.

## Figures and Tables

**Figure 1 ncrna-10-00032-f001:**
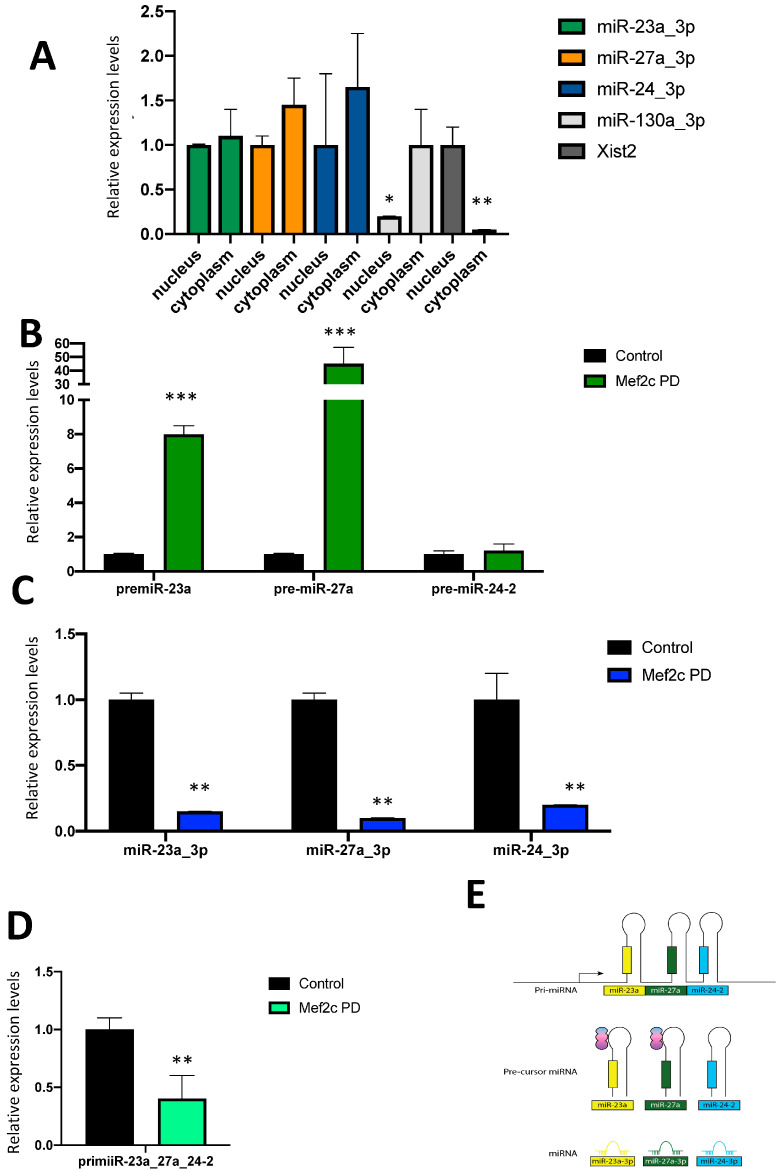
Panel (**A**) RT-qPCR analyses of the nuclear and cytoplasmic distribution of miR-23a_3p, miR-27a_3p and miR-24_3p mature microRNAs in HL1 cardiomyocytes. Note that all three microRNAs are similarly expressed in the nucleus and cytoplasm in contrast to miR-130a, which is primarily cytoplasmic, and the long non-coding RNA Xist2, which is preferentially nuclear. Panel (**B**) RT-qPCR analyses of Mef2c pulldown assays for pre-miR-23a, pre-miR-27a and pre-miR-24, respectively. Note that increased levels are observed for pre-miR-23a and pre-miR-27a but not for pre-miR-24. Panel (**C**) RT-qPCR analyses of Mef2c pulldown assays for mature miR-23a_3p, miR-27a_3p and miR-24_3p, respectively. Note that none of the mature microRNAs are increased after Mef2c pulldown assays. Panel (**D**) RT-qPCR analyses of Mef2c pulldown assays for pri-miR-23-miR-27a-miR-24-2. Panel (**E**) Schematic representation of the Mef2c association with the miR-23a-miR-27a-miR-24-2 clustered microRNAs. All data are normalized to Gapdh for mRNA expression analyses and to 5S for microRNA expression analyses. * *p* < 0.05, ** *p* < 0.01, *** *p* < 0.001.

**Figure 2 ncrna-10-00032-f002:**
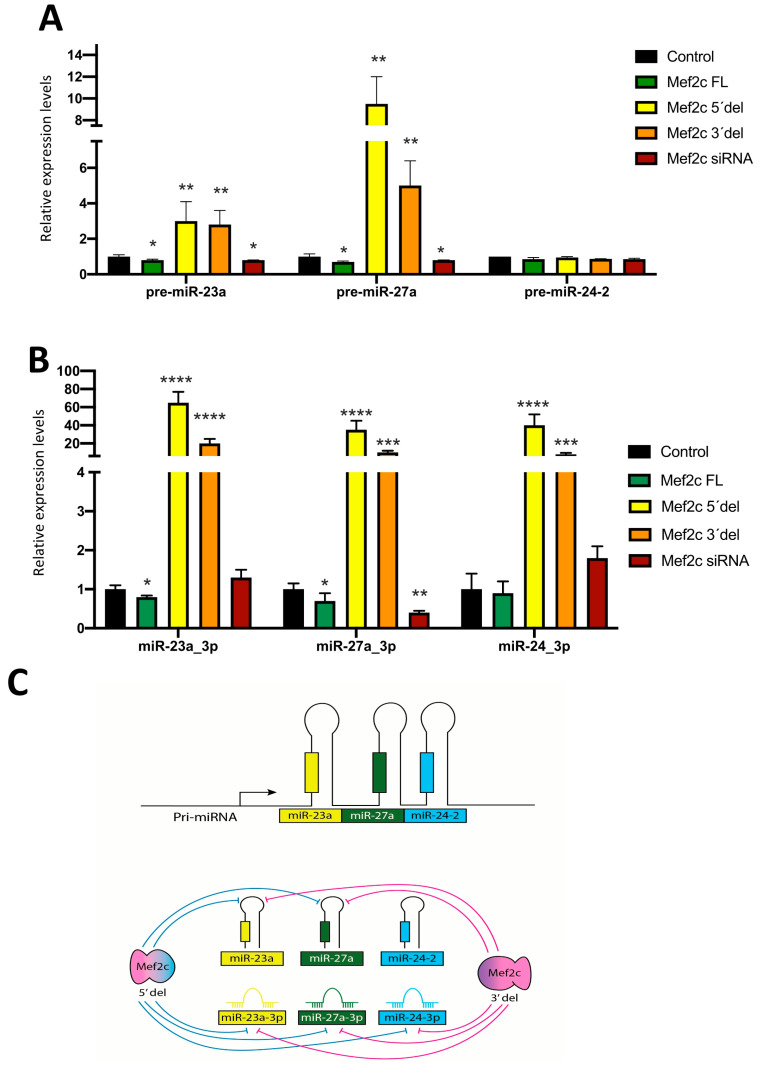
Panel (**A**) RT-qPCR analyses of pre-miR-23a, pre-miR-27a and pre-miR-24 expression after over-expression of Mef2c full-length (FL), Mef2c 5′del, Mef2c 3′del and Mef2c siRNA in HL1 cardiomyocytes, respectively. Note increased levels are observed for pre-miR-23a and pre-miR-27a but not for pre-miR-24 after Mef2c 5′del and Mef2c 3′del over-expression, while Mef2c FL and Mef2c siRNA significantly decreased them. Panel (**B**) RT-qPCR analyses of mature miR-23a_3p, miR-27a_3p and miR-24_3p expression after over-expression of Mef2c full-length (FL), Mef2c 5′del, Mef2c 3′del and Mef2c siRNA in HL1 cardiomyocytes, respectively. Note increased levels are observed for all mature microRNAs after Mef2c 5′del and Mef2c 3′del over-expression, while Mef2c FL significantly decreased miR-23a_3p and miR-27a_3p but not miR-24, while Mef2c siRNA only decreased miR-27a_3p. Panel (**C**) Schematic representation of the Mef2c 5′del and Mef2c 3′del regulation of the miR-23a-miR-27a-miR-24-2 clustered microRNAs. All data are normalized to Gapdh for mRNA expression analyses and to 5S for microRNA expression analyses. * *p* < 0.05, ** *p* < 0.01, *** *p* < 0.001, **** *p* < 0.0001.

**Figure 3 ncrna-10-00032-f003:**
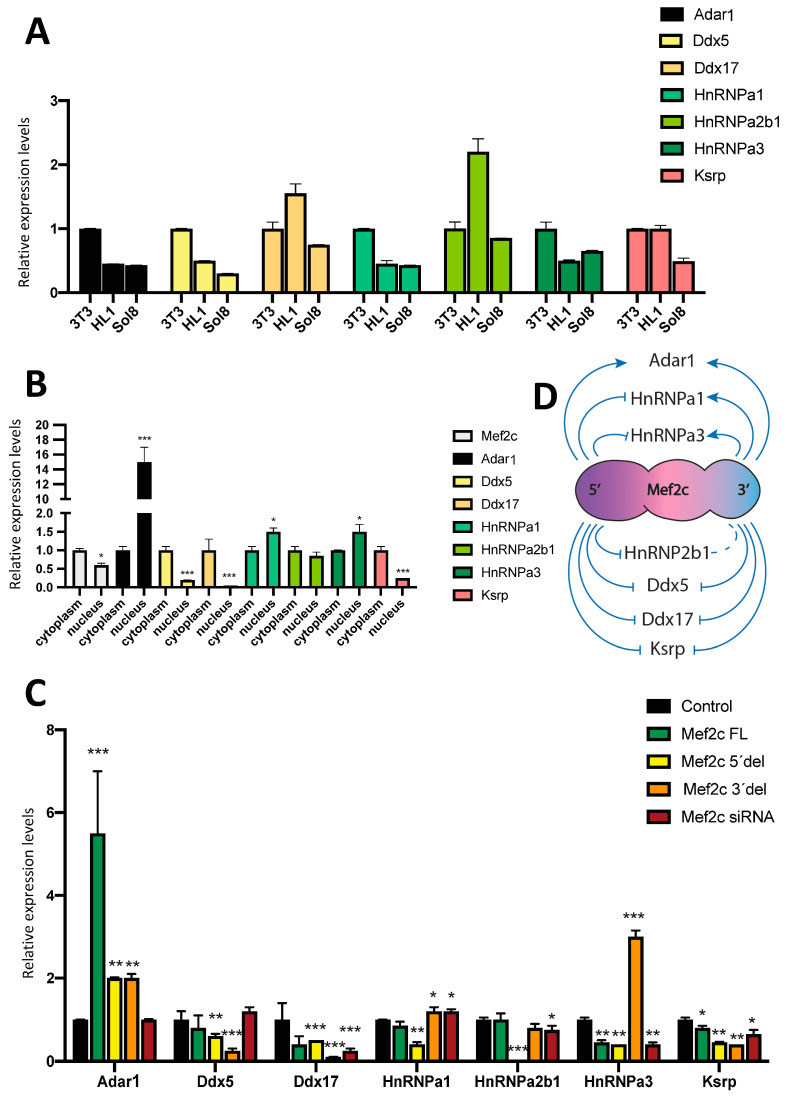
Panel (**A**) RT-qPCR analyses of distinct RNPs (Adar1, Ddx5, Ddx17, HnRNPa1, HnRNPa2b1, HnRNPa3 and Ksrp) in three distinct cell lines: 3T3 fibroblasts, HL1 cardiomyocytes and Sol8 skeletal muscle myoblasts. Observe that these RNPs display distinct expression levels on each of the tested cell lines. Panel (**B**) RT-qPCR analyses of the nuclear and cytoplasmic distribution of these RNPs in HL1 cardiomyocytes. Note that Adar1 is preferentially expressed in the nucleus, while Ddx5, Ddx17 and Ksrp are preferentially expressed in the cytoplasm. Panel (**C**) RT-qPCR analyses of RNP expression after over-expression of Mef2c full-length (FL), Mef2c 5′del, Mef2c 3′del and Mef2c siRNA in HL1 cardiomyocytes, respectively. Note that these RNPs are distinctly regulated by each of the Mef2c constructs analyzed. Panel (**D**) Schematic representation of the Mef2c 5′del and Mef2c 3′del regulation of the RNPs. All data are normalized to Gapdh expression. * *p* < 0.05, ** *p* < 0.01, *** *p* < 0.001.

**Figure 4 ncrna-10-00032-f004:**
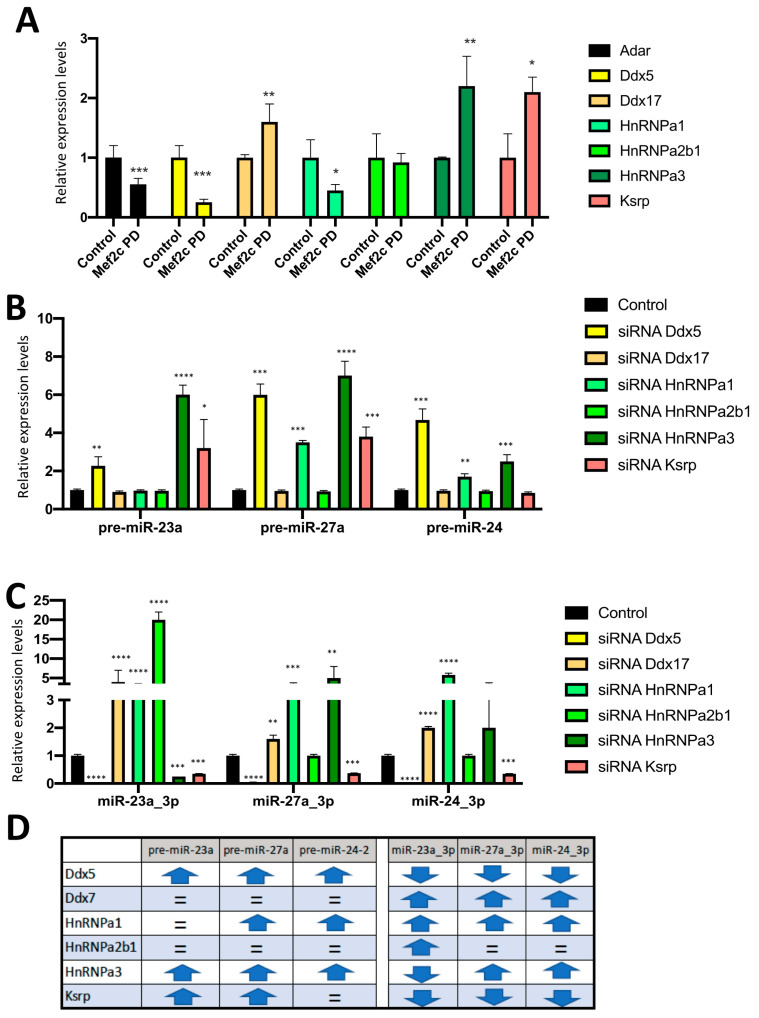
Panel (**A**) RT-qPCR analyses of Mef2c pulldown assays for RNPs in HL1 cardiomyocytes. Note that increased levels for Ddx17, HnRNPa3 and Ksrp are observed. Panel (**B**) RT-qPCR analyses of pre-miR-23a, pre-miR-27a and pre-miR-24-2 expression after silencing each of the RNPs previously tested in HL1 cardiomyocytes, respectively. Note that silencing Ddx5 and HnRNPa3 enhanced the expression of all pre-microRNAs while silencing Ksrp only upregulated pre-miR-23a and pre-miR-27a but not pre-miR-24-2. Panel (**C**) RT-qPCR analyses of mature miR-23a_3p, miR-27a_3p and miR-24_3p expression after silencing each of the RNPs previously tested in HL1 cardiomyocytes, respectively. Note that silencing Ddx5 and Ksrp decreased the expression of all pre-microRNAs while silencing HnRNPa2b1 selectively upregulated only pre-miR-23a. Panel (**D**) Schematic representation of the effects of RNP silencing on miR-23a-miR-27a-miR-24-2 pre-miRNA and mature microRNA expression, respectively. All data are normalized to Gaped for mRNA expression analyses and to 5S for microRNA expression analyses. * *p* < 0.05, ** *p* < 0.01, *** *p* < 0.001, *** *p* < 0.0001.

**Figure 5 ncrna-10-00032-f005:**
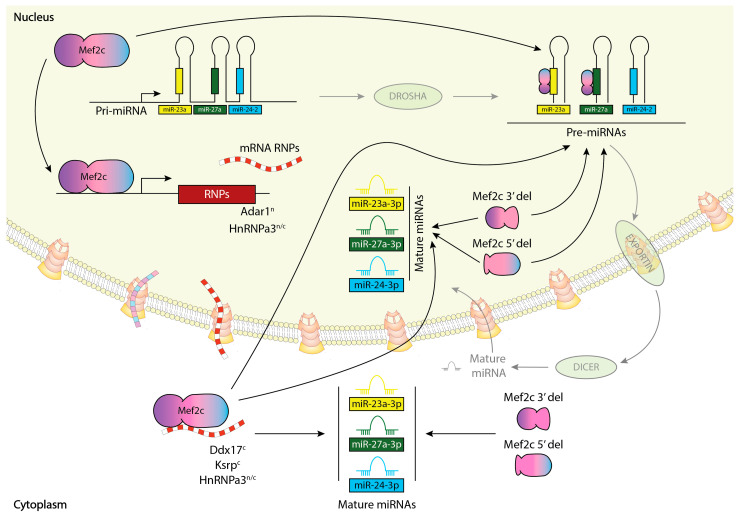
Schematic representation of Mef2c post-transcriptional regulation of miR-23a-miR-27a-miR-24-2 clustered pre-miRNAs and microRNAs, respectively.

## Data Availability

Available upon request to the corresponding author.

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
