# Peer review of "MEF2C Directly Interacts with Pre-miRNAs and Distinct RNPs to Post-Transcriptionally Regulate miR-23a-miR-27a-miR-24-2 microRNA Cluster Member Expression"

_ncrna, 2024, doi:10.3390/ncrna10030032_

Round 1
Reviewer 1 Report
Comments and Suggestions for Authors
MEF2C is one of the transcription factors involved in myogenesis. It has been shown to regulate expression of miR23a-miR27a-miR24-2 microRNA cluster at the level of transcription. In this manuscript Lozano-Velasco et al., report additional role of MEF2C in expression of miR23a-miR27a-miR24-2 microRNA cluster at the post-transcriptional level. This conclusion is primarily based on the physical association of MEF2C with pre-miR23a and pre-miR27a, nut not with the primary or mature transcripts of these two miRNAs. The additional experimental data provided by authors in support of their post-transcriptional regulation hypothesis is weak. The study, at best, can be described as preliminary and inconclusive. Furthermore, at multiple places in the manuscript either data is not there in the referred figure or conclusion is not supported by data in the referred figure.
1. Authors performed overexpression and silencing of MEF2C. No evidence is provided that overexpression and silencing resulted in enhanced expression and a decrease in the level of MEF2C in the cell. This data must be there in supplementary file.
2. Authors investigated the effect of MEF2C-5’ deletion and MEF2C-3’ deletion on expression of miR23a-miR27a-miR24-2 cluster. Authors mention that FL, 5’ deletion mutant and 3’ deletion mutant of MEF2C were expressed at the similar level in cells. They referred to the Supplementary Fig 1C while describing this result. There is however no WB showing that MEF2C FL, 5’ deletion mutant and 3’ deletion mutant were expressed at the similar level in cells.
3. Supplementary Fig 1D is referred to while describing that FL and 3’ deletion constructs were able to activate transcription of miR23a-miR27a-miR24-2 cluster, but 5’ deletion mutant could not. This data is not there in Supplementary Fig 1D.
4. Authors state that overexpression of FL MEF2C and silencing resulted in minimal but significant downregulation of pre-miR23a and pre-miR27a. Fig 2A, however does not support the author’s conclusion and there is hardly any decrease in the level of these two miRNA upon overexpression or silencing of FL MEF2C.
5. No explanation is provided for increase in level of mature miR23a and miR24-2 RNA upon silencing of MEF2C in Fig. 2B.
6. It is not clear how post-transcription regulation of pre-miR-23a and pre-miR27a by MEF2C can be explained in terms of MEF2C-mediated regulation of RNPs.
Comments on the Quality of English LanguageMinor edits are required.
Author Response
First of all, we would like to thank the reviewer for his/her comments on our study that has allowed us to improved it. We apologize since we noticed indeed that part of the Supplementary Figures were not properly uploaded and thus important information was missing that would certainly help to clarify our conclusions.
- Authors performed overexpression and silencing of MEF2C. No evidence is provided that overexpression and silencing resulted in enhanced expression and a decrease in the level of MEF2C in the cell. This data must be there in supplementary file.
First of all, we would like to apologize since Supplementary Figure 1 was not properly uploaded (it corresponded to an old version) and thus important information was missing. Within the updated version of Supplementary Figure 1 (panel C) we provide evidence of the overexpression and silencing of MEF2C at transcriptional level, respectively. Additionally, and following the recommendation of the reviewer we now provide evidence that MEF2C overexpression and silencing resulted, as expected, on increased and decreased MEF2C expression at protein level, respectively (Supplementary Figure 1D).
- Authors investigated the effect of MEF2C-5’ deletion and MEF2C-3’ deletion on expression of miR23a-miR27a-miR24-2 cluster. Authors mention that FL, 5’ deletion mutant and 3’ deletion mutant of MEF2C were expressed at the similar level in cells. They referred to the Supplementary Fig 1C while describing this result. There is however no WB showing that MEF2C FL, 5’ deletion mutant and 3’ deletion mutant were expressed at the similar level in cells.
As previously said, we are very sorry for the mistake uploading Supplementary Figure 1. Following the recommendation of the reviewer we now provide evidence that FL, 5’ deletion mutant and 3’ deletion mutant of MEF2C are expressed at the similar level in cells at transcript level (Supplementary Figure 1C) and additionally we now provide also evidences at protein level (Supplementary Figure 1D).
- Supplementary Fig 1D is referred to while describing that FL and 3’ deletion constructs were able to activate transcription of miR23a-miR27a-miR24-2 cluster, but 5’ deletion mutant could not. This data is not there in Supplementary Fig 1D.
As previously said, we are very sorry for the mistake uploading Supplementary Figure 1. In the revised version of the manuscript, Supplementary Figure 1E provides evidence of the distinct transcriptional transactivation potential of the different MEF2C constructs, i.e. FL, MEF2C 5´del and MEF2C 3´del on the L construct of the miR-23a-miR-27a-miR-24-2 locus, respectively.
- Authors state that overexpression of FL MEF2C and silencing resulted in minimal but
significant downregulation of pre-miR23a and pre-miR27a. Fig 2A, however does not support the author’s conclusion and there is hardly any decrease in the level of these two miRNA upon overexpression or silencing of FL MEF2C.
We respectfully disagree with the statements provided by the reviewer. Figure 2A demonstrates that overexpression of MEF2C FL resulted in a small (i.e. approximately 20%), but significant downregulation of premiR-23a and pre-miR-27a but not pre-miR-24-2. Similarly, overexpression of MEF2C FL resulted in a significant downregulation of mature miR-23a and miR-27a but not miR-24. Thus, these data support the notion that differential regulation of miR-23a-miR-27a-miR-24-2 cluster members is exerted at pre-miRNA level, as stated in our manuscript.
- No explanation is provided for increase in level of mature miR23a and miR24-2 RNA upon silencing of MEF2C in Fig. 2B.
We respectfully disagree with the statements provided by the reviewer. As illustrated in Figure 2B, there no significant difference in the level of expression of miR23a and miR24-2 RNAs upon silencing of MEF2C, respectively and thus this is the reason why no explanation is provided.
- It is not clear how post-transcription regulation of pre-miR-23a and pre-miR27a by MEF2C can be explained in terms of MEF2C-mediated regulation of RNPs.
In this study we provided evidence that MEF2C can directly interact with pre-miR-23a and pre-miR27a but not pre-miR-24-2. Additionally, we provided evidences that MEF2C can physically interact with several RNPs at transcript level as well as modulate their expression. Finally, we also provide evidences that silencing of different RNPs can modulate the expression of pre-miR-23a, pre-miR27a and pre-miR-24-2 as well as their mature forms. Therefore, our data demonstrate that MEF2C can influence mature miR-23a and miR-27a either by directly interacting with their premiRNAs, or indirectly by interacting with distinct RNP transcripts, that eventually leads to distinct RNP proteins that influence the mature miR-23a and miR-27a output. Surprisingly, silencing of none of these RNPs can recapitulate the MEF2C modulation of the mature miR-23a, miR-27a and miR-24 supporting the notion that combinatorial rather than the action of a single RNP is required to modulate their expression.
Reviewer 2 Report
Comments and Suggestions for Authors
In this manuscript, Estefania Lozano-Velasco and colleagues have studied the complex role of Myocyte enhancer factor 2C (MEF2C), which has been shown as a transcription factor, in regulating gene expression via transactivation of the long regulatory element upstream of 'miR-23a-miR-27a-miR-24-2 cluster' transcriptional start site.
The authors have studied how MEF2C is involved in the expression and post transcriptional regulation of this microRNA cluster and which domains of this protein is involved. Additionally, they study the interactions of MEF2C with pre-miRNAs and RNPs like Ksrp, HnRNPa3, and Ddx17.
The hypothesis behind these complex and hidden patterns of miRs expression regulation is interesting and some parts of the experiments support the idea. However, the whole experimental setup is lacking connection and in some cases not supporting enough of what has been concluded and I would recommend those sections to be rephrased and additional experiments be performed to address the questions in place!
Below are some recommendations and technical concerns:
-
Following a previous work showing that MEF2C over-expression and silencing, respectively, displayed distinct effects on each of the miR-23a-miR-27a-miR-24-2 cluster members, without affecting pri-miRNA expression levels in different cell types.
-
For validation of the cell fractionation experiment there is a need to do a Western Blot for example using Histon and Tubulin antibodies as nuclear and cytoplasmic enriched proteins respectively. Also for the qPCR assay, the better control transcript for a microRNA study is a RNA species close to the size of pre-miRs such as U6 etc. and using Xist which is a long RNA is not a proper candidate.
-
Since qPCR is the main validation assays through this project, and PCR amplification of the pre-microRNAs is a difficult experiment design due to the structural complexity, there is a need to do Sanger sequencing for the qPCR products of each pre-microRNA. Additionally, all details about the kits and methods and sequence of the primers used, should be provided.
-
In addition to the qPCR, and to overcome any structural complexity of the pre-miRs for qPCR, I would strongly recommend confirming the qPCR assays by doing Northern Blot for each pre-MicroRNA for all the conditions. This Blotting is important for the final conclusion and if MEF2C is binding to any pre-miRs or not!?
-
It is not clear which tool has been used for the prediction of MEF2C and pre-miRs interaction? Also fig-2C is misleading the reader if the mature microRNAs are 5p or 3p derived!?
-
Regarding Fig-2, while I find the 5' and 3' deletion assay interesting for follow up, it is clear that neither overexpression nor siRNA KD of the MEF2C has not changed the level of any of these miRs in their mature format when compared to the control. In the case of siRNA KD, considering the protein half-life is important, to see its effect in the experiment afterward if any! Therefore, a better explanation need to be provided or more focus need to be done on the 3' and 5' domain deletion effect on the fine tuning not only that microRNA cluster, but also some other microRNAs as control in case these deletions have significant effect on the biogenesis of other microRNAs as well.
-
Generally, the study of MEF2C interaction with other RNPs is interesting, however it is important to address each question by propper assay. A qPCR assay is for the study of MEF2C variants on the expression of those RNPs at the RNA level, while for real effect on RNPs expression level and also physical interacting study there is a need for antibody-based WB assays with and without MEF2C pulldown. Additionally, there is a need to propose a stronger scenario for connecting the RNPs-MEF2C-miR23a-27a-24 cluster story.
-
Authors need to be consistent with writing the name of the genes and proteins. For example for pulling down a protein it should be MEF2C all with capital letters and if they mean the gene or RNA, for mouse it should be Mef2c in italic.
Minor grammar corrections is needed.
Author Response
First of all, we would like to thank the reviewer for his/her comments on our study that has allowed us to improved it. Following the recommendation of the reviewer, we provided additional data and we have partly rephrased the results in the revised version of the manuscript.
Below are some recommendations and technical concerns:
- Following a previous work showing that MEF2C over-expression and silencing,
respectively, displayed distinct effects on each of the miR-23a-miR-27a-miR-24-2
cluster members, without affecting pri-miRNA expression levels in different cell types.
The reviewer is right. We have previously reported that MEF2C can distinctly modulate mature miR-23a-miR-27a-miR-24-2 cluster members without altering the expression of miR-23a-miR-27a-miR-24-2 primiRNA (Hernandez-Torres et al., 2014)
- For validation of the cell fractionation experiment there is a need to do a Western Blot for example using Histone and Tubulin antibodies as nuclear and cytoplasmic enriched proteins respectively. Also for the qPCR assay, the better control transcript for a microRNA study is a RNA species close to the size of pre-miRs such as U6 etc. and
using Xist which is a long RNA is not a proper candidate.
The reviewer suggest that enrichment of cell fractionation should be performed by Western blot analyses using for example nuclearly enriched proteins such as histones and cytoplasmic enriched proteins such as tubulin since he/she considers that using Xist lncRNA is not a proper candidate. We respectfully disagree since monitoring protein distribution in nuclear and cytoplasmic fractions would necessarily needed to be done in distinct cell fractions, as the same fraction (either nuclear or cytoplasmic) cannot be used to yield both protein and RNA simultaneously. Using pre-miRNA U6 does not warrant either enrichment on nuclear/cytoplasmic fraction as it has been reported that microRNAs can be in both nuclear/cytoplasmic compartment (Zhang et al., 2019; Catalanotto et al. 2016). We therefore opted to used Xist as it has been previously reported to be nuclearly restricted (Obuse & Hirose, 2023)
Zhang X, Shen B, Cui Y. Ago HITS-CLIP expands microRNA-mRNA interactions in nucleus and cytoplasm of gastric cancer cells. BMC Cancer. 2019 Jan 8;19(1):29. doi: 10.1186/s12885-018-5246-0. PMID: 30621629; PMCID: PMC6325853.
Catalanotto C, Cogoni C, Zardo G. MicroRNA in Control of Gene Expression: An Overview of Nuclear Functions. Int J Mol Sci. 2016 Oct 13;17(10):1712. doi: 10.3390/ijms17101712. PMID: 27754357; PMCID: PMC5085744.
Obuse C, Hirose T. Functional domains of nuclear long noncoding RNAs: Insights into gene regulation and intracellular architecture. Curr Opin Cell Biol. 2023 Dec;85:102250. doi: 10.1016/j.ceb.2023.102250. Epub 2023 Oct 6. PMID: 37806294.
- Since qPCR is the main validation assays through this project, and PCR amplification
of the pre-microRNAs is a difficult experiment design due to the structural complexity,
there is a need to do Sanger sequencing for the qPCR products of each pre-microRNA.
Additionally, all details about the kits and methods and sequence of the
primers used, should be provided.
Following the recommendation of the reviewer we have cloned the PCR amplification products resulting from qPCR analyses in different experimental conditions and cloned into T cloning vector. Subsequently Sanger sequencing was performed and demonstrating that in all cases qPCR amplification yielded to the expected pre-miRNA sequence (Supplementary Figure 2B). Additionally, as suggested by the reviewer, details of the kits, sequencing and primers used is provided in the revised version of the manuscript.
- In addition to the qPCR, and to overcome any structural complexity of the pre-miRs
for qPCR, I would strongly recommend confirming the qPCR assays by doing
Northern Blot for each pre-MicroRNA for all the conditions. This Blotting is important
for the final conclusion and if MEF2C is binding to any pre-miRs or not!?
We respectfully disagree with the recommendation of doing Northern blot for the qPCR amplification fragments since we provided undoubtedly evidence by PCR amplification, cloning and sequencing that demonstrate in all cases that qPCR yielded the expected pre-miRNA amplicon, as reported in Supplementary Figure 2A & 2B, supporting the notion that structural complexity of the miR-23a, miR-27a and miR-24 premiRNAs does not hinder appropriate qPCR amplification.
Reviewer 3 Report
Comments and Suggestions for Authors
In the manuscript by Lozano-Velasco et al., the authors found that MEF2C activates a specific long regulatory element upstream of miR-23a-miR-27a-miR-24-2, yielding distinct effects on individual mature cluster members without altering pri-miRNA levels, indicating additional MEF2C-driven regulatory mechanisms. The manuscript is well-written and structured, besides one minor comment that the authors should use specific y-axis labels for all bar graphs but not the confusing “arbitrary units”.
Author Response
First of all, we would like to thank the reviewer for his/her comments on our study that has allowed us to improved it. Following the recommendation of the reviewer, we have modified the y-axis, avoiding the use of the term "arbitrary units".
Round 2
Reviewer 1 Report
Comments and Suggestions for Authors
The missing figures in supplementary file are appropriate and corroborates author's conclusions.
Comments on the Quality of English LanguageOnly minor editing is required.
Author Response
We would like to thank the reviewer his/her critical comments that has allowed us to improved our manuscript. Following the recommendation of the reviewer, we have carefully edited the revised version of the manuscript to avoid plausible grammatical mistakes.
Reviewer 2 Report
Comments and Suggestions for Authors
In their reply to comment 2, there are some misunderstanding. First, it is possible to harvest both RNA and protein from the same sample following routine cell fractionation and RNA/protein extraction. If difficult to manage then two fractionation extraction could be done per sample at the same, one for RNA and one for protein. Afterward they could do WB for both Histone and Tubuline on both fractions and strong band should show up for Histone in the nuclear fraction and for Tubuline in cytoplasm and vise versa.
Additionally, the authors mistakenly think U6 RNA is microRNA (pre-miR) which is not, and it is nuclear specific small RNA and frequently has been used for such controls!
Regarding my previous comment 4, again authors have not read my suggestion carefully that recommended to do Northern Blot to confirm the presence of the claimed pre-miRs they have done by qPCR only . This is more reliable than qPCR alone. NB is being done on RNA and not PCR product which is DNA. These simple mistakes are not expected from a scientist doing and writing such a research paper. However, if the authors have sequenced the qPCR products and they have not seen extra unspecific bands at least by running on a gel, that can replace the NB and is acceptable.
Comments on the Quality of English Language
Minor grammar corrections is needed.
Author Response
In their reply to comment 2, there are some misunderstanding. First, it is possible to harvest both RNA and protein from the same sample following routine cell fractionation and RNA/protein extraction. If difficult to manage then two fractionation extraction could be done per sample at the same, one for RNA and one for protein. Afterward they could do WB for both Histone and Tubuline on both fractions and strong band should show up for Histone in the nuclear fraction and for Tubuline in cytoplasm and vice versa.
First of all, we are very sorry for the misunderstanding provided on our reply to comment 2. We agree with the reviewer that it is plausible to collect both RNA and protein from the sample. Yet in our hands, this approach has always yielded to an enriched bias to RNA, providing very little protein yield. This outcome has been observed when using a modified version of Trizol protocol (2.5 volumes of ethanol to 1 volume of protein solution ‘phenol-phase’) or using acetone for protein precipitation after flow through collection during RNA isolation with RNAeasy mini kit. We also agree with the reviewer that separating the samples in two halves and processing one for RNA and the other for protein might be an alternative, yet in this case they are not exactly the same samples. We therefore have used alternative methods to Western blot for discerning nuclear vs cytoplasmic enriched based on RNA detection.
Additionally, the authors mistakenly think U6 RNA is microRNA (pre-miR) which is not, and it is nuclear specific small RNA and frequently has been used for such controls.
We are again sorry for the misunderstanding. We are aware that U6 RNA is not a microRNA, but a non-coding small nuclear RNA. We agree with the reviewer indeed that such non-coding small nuclear RNA (U6 RNA) has been frequently used as control, similarly as 5S ribosomal RNA, and this is the reason why we used it.
Regarding my previous comment 4, again authors have not read my suggestion carefully that recommended to do Northern Blot to confirm the presence of the claimed pre-miRs they have done by qPCR only . This is more reliable than qPCR alone. NB is being done on RNA and not PCR product which is DNA. These simple mistakes are not expected from a scientist doing and writing such a research paper. However, if the authors have sequenced the qPCR products and they have not seen extra unspecific bands at least by running on a gel, that can replace the NB and is acceptable.
We apologize again for the misunderstanding. We are fully aware that Northern blots are done upon RNA and can directly detect the presence of RNA molecules, while RT-qPCR results in a DNA product, yet emanating from RNA template by reverse transcriptase conversion into complementary DNA. We are indeed very happy to hear that our sequencing results of the resulting amplification products, together with the evidence of single bands on the gel can replace the requested Northern blot analyses, making them acceptable to the reviewer.